# Determination and Risk Assessment of Flavor Components in Flavored Milk

**DOI:** 10.3390/foods12112151

**Published:** 2023-05-26

**Authors:** Baorong Chen, Xiaodan Wang, Yumeng Zhang, Wenyuan Zhang, Xiaoyang Pang, Shuwen Zhang, Jing Lu, Jiaping Lv

**Affiliations:** 1Institute of Food Science and Technology, Chinese Academy of Agricultural Sciences, Beijing 100081, Chinalvjiapingcaas@126.com (J.L.); 2Beijing Advanced Innovation Center for Food Nutrition and Human Health, Beijing Engineering and Technology Research Center for Food Additives, School of Food and Health, Beijing Technology and Business University, Beijing 100048, China

**Keywords:** flavored milk, SPME-GC/MS, flavor components, acceptable daily intake, toxicological concern threshold

## Abstract

This study aimed to determine chemical composition and assess exposure in flavored milk among Chinese residents, based on risk assessment methodologies of acceptable daily intake (ADI) and toxicological concern threshold (TTC). Esters (32.17%), alcohols (11.19%), olefins (9.09%), aldehydes (8.39%), and ketones (7.34%) comprised the majority of the flavoring samples. Methyl palmitate (90.91%), ethyl butyrate (81.82%), and dipentene (81.82%) had the highest detection rates in flavor samples. This study screened fifteen flavor components of concern and discovered that 2,3,5-trimethylpyrazine, furfural, benzaldehyde, and benzenemethanol were detected in 100% of flavored milk samples. Benzenemethanol was found in the highest concentration (14,995.44 μg kg^−1^). The risk assessment results revealed that there was no risk for Chinese residents in consuming flavored milk, and the maximum per capita daily consumption of 2,3,5-trimethylpyrazine, furfural, and benzenemethanol were 226.208 g, 140.610 g, and 120.036 g, respectively. This study could provide guidelines for amounts of flavor additive ingredients in milk.

## 1. Introduction

According to the Chinese Dairy Industry Quality Report (2019), ultra-high temperature sterilized milk, flavored milk, fermented milk, and pasteurized milk accounted for 40.6%, 28.1%, 21.3%, and 10% of total liquid milk consumption, respectively. Flavored milk is a type of sterilized liquid milk that comprises at least 80% raw bovine (caprine) milk or reconstituted milk (GB 25191-2010), and serves as a nutritional alternative to plain milk [1]. Distribution of volatile compounds is directly related to food flavor [2]. Flavor can be added in appropriate proportions and unlimited amounts according to GB 2760-2014 “National Standard for Food Safety Food Additive Use”. However, maltol directly stimulates Cyp1a1 gene expression [3]. Maltol has harmful effects on the skin, eyes, and respiratory system [4]. Mutagenesis and genotoxicity are linked to furfural compounds [5]. Moreover, the safety of flavor components is frequently neglected, with the characteristic of self-limiting and low dosage.

The Flavor and Extract Manufacturers Association (FEMA) has clearly stated the safe amount of flavor additives in soft drinks, candies, baked goods, puddings, and meat, as shown in Table 1. Unfortunately, the amount added to dairy products was rarely mentioned. Flavored dairy products, as the most popular dairy products among children, should be given greater attention. Acceptable daily intake (ADI) was initiated by the Joint FAO/WHO Expert Committee on Food Additives (JECFA) in 1961 and is accessible for toxicological evaluation [6]. When estimated daily intake (EDI) is smaller than ADI, it does not cause harm [7]. Toxicological concern threshold (TTC) is also a useful screening and prioritizing measure for assessing food safety [8]. Each substance is examined and categorized based on chemical structure, and divided into three human exposure thresholds (1800, 540 and 90 μg p^−1^ d^−1^). When a substance’s human exposure is lower than the threshold value, the potential safety risk is negligible. Risk exposure assessment needs to combine the concentration of chemicals and the amount of food consumed [9]. A common means of determining this is to interview and record by questionnaire, such as investigating and analyzing the correlation between dairy product consumption and cardiovascular diseases [10], serum vitamin D deficiency [11], ACEN [12], cultural factors and purchasing behavior [13]. Hence, it is an excellent choice that this study adopts a questionnaire method to investigate the consumption of flavored milk in different age groups.

Currently, flavor extraction and determination procedures in dairy products include solid phase micro-extraction (SPME) [14,15], supercritical CO_2_ fluid extraction (SFE), and dynamic headspace (DHS) [16]. The flavor compounds of infant milk powder [17], reduced-fat dairy products [18], raw goat cheese [19], pea protein beverages [20], camel milk [21], cattle and sheep milk [22], mascarpone cheese [23], and sweet condensed milk [24] were mainly determined by gas chromatography–mass spectrometry (GC/MS). We investigated and measured the consumption of flavored milk in various age groups and established a risk assessment model for flavored milk. This risk assessment has been carried out by taking the mean value as the food consumption data and the maximum detected value as the substance concentration data. This study was necessary and important to provide additive standards for flavor ingredient amounts in milk.

## 2. Materials and Methods

### 2.1. Reagents and Equipment

The online survey “Residents’ flavored milk consumption questionnaire” was created on the Questionnaire website and was available for completion from 1 January 2021 to 30 May 2021. Appendix A has provided details of the questionnaire model. Flavor samples of five brands (Givaudan, MANE, Huacheng, ARTSCI, and Firmenich) were collected from Chinese dairy companies. Twenty-eight flavored milk samples were obtained from Chinese supermarkets, including brands such as Bright Dairy, Mengniu Dairy, Yili Dairy, and Sanyuan Daiy. 2,3,5-trimethylpyrazine (99%), phenol, 2-methoxy-4-(2-propenyl)- (>99.5%), maltol (99%), benzenemethanol (≥99.5%), methyl salicylate (≥99.5%), benzyl acetate (≥99.7%), 5-methylfurfural (98%), linalool (98%), benzaldehyde (≥99.5%), furfural (≥99.5%), hexanoic acid 2-propenyl ester (98%), 1-hexanol (>99.5%), ethyl 3-methylbutyrate (≥99.7%), and 2-methylpropanal (99%) were purchased from Shanghai Aladdin Biochemical Technology Company (Shanghai, China). Methyleugenol (≥98%) was purchased from Shanghai Yuanye Biology Science and Technology Company (Shanghai, China). Methanol (99.9%) was collected from Thermofisher Scientific Technology Company (Boston, MA, USA). C7-C40 N-alkanes (99.5%) was purchased from an American company, O2si (USA). 50/30 μm divinylbenzene carbon molecular sieve polydimethylsiloxane (DVB/CAR/PDMS) extraction head was obtained from the American company Supelco (Missouri, USA). Gas chromatography–mass spectrometry equipment was purchased from Thermo Scientific. A DB-WAX column (60 m × 0.25 mm × 0.25 μm) was obtained from Agilent (Santa Clara, CA, USA).

### 2.2. Sample Pretreatment

The method of measuring flavor samples was that used by Li Ning et al. [21]. Fifteen flavor components of concern were configured into mixed standard solutions and stored in a 4 °C refrigerator. An external standard approach was used for quantitative analysis. Flavored milk samples were put in solid phase extraction bottles for investigation. The 50/30 μm DVB/CAR/PDMS fiber head was aged at 250 °C until the baseline was stable [25]. The extraction temperature of the solid phase micro extraction platform was set to 60 °C and rotation speed to 800 rpm, then balanced for 10 min after inserting the aged fiber extraction head. A distance of 1.5 cm was kept between the fiber head and the liquid surface. The extraction temperature was kept at 55 °C for 50 min, and the extraction fiber head was put into the GC–MS injection port at 250 °C for 5 min of analysis.

### 2.3. GC/MS Conditions

A DB-WAX column was adopted for GC separation. The heating procedure was as follows: the initial temperature was 40 °C for 5 min, and the temperature rose to 150 °C at 3 °C/min, then to 230 °C at 6 °C/min and held for 5 min. The inlet temperature was set to 250 °C and flow rate set to at 1.0 mL/min. No shunt injections and solvent was delayed for 3 min. MS settings were as follows: electron ion source (EI), no solvent delay, SCAN mode, mass scanning range *m*/*z* 35~450 u. NIST, Wiley 9, and other libraries were searched for flavor compounds. Those compounds with an SI above 750 and Total Score above 90 were taken as preliminary screening results. Retention Index (RI), as calculated by the instruments, with RI obtained from the retrieval database, was used to further determine flavor compounds.

### 2.4. Exposure Risk Assessment Methods in Flavored Milk

The estimated daily intake (EDI) and per capita daily intake (PCI) of flavor components of concern in flavored milk were calculated according to Equations (1) and (2).
EDI (μg/(kg·bw)/day) = F × C/W.(1)
PCI (μg/person/day) = F × C.(2)
in which F is daily intake of dairy products per capita in kg/person/day; C is maximum flavor additive content in μg kg^−1^; and W is average weight in kg.

### 2.5. Data Analysis

The data was shown by mean ± SD. Non-parametric tests were used to analyze significant differences by IBM SPSS Statistics 26. The bar plot diagram, heat map, and principal component analysis were drawn in Tutools (https://www.cloudtutu.com/ accessed on 9 March 2023).

## 3. Results and Discussion

### 3.1. Distribution of Sample Characteristics

There were 2108 valid questionnaires obtained, with a questionnaire efficiency of 90.36%. The sociological characteristics of participants are shown in Appendix A, including genders, ages, occupations, region of residence, and flavor preferences. In this study, the distribution of gender was lopsided, with females (60.2%) more represented than males (39.8%). This gender ratio is consistent with the study investigating the impact of society and lifestyle factors on dairy consumption [26]. The average weight (kg) and daily consumption per capita (g) from different ages and regions were shown in Table 2. There was a significant difference in the daily per capita intake of flavored milk between males and females (*p* < 0.01). Males consumed more than females among all age groups, but the results were not consistent with Marek Kardas et al. [27]. Teenagers (<18 years old) had the highest dairy consumption (males 77.57 ± 89.34 g, females 55.14 ± 52.44 g), followed by persons aged 18~24 years (males 57.66 ± 89.59 g, females 50.57 ± 56.66 g). Previous studies indicated teenagers consumed more flavored milk than plain milk in order to enhance their consumption of sugar and fat [28]. Chinese teenagers (66 g) consumed more flavored milk per day than adolescents (50 g) in the United States [1].

This study also analyzed flavored milk consumption in seven regions, namely eastern, southern, central, northern, northwestern, southwestern, and northeastern China. Males from northern China (55.13 ± 133.73 g) and northeastern China (64.33 ± 87.32 g) had higher flavored milk consumption than other regions. In the northeast, the consumption of males (64.33 ± 87.32 g) was much higher than that of females (36.12 ± 48.83 g). Gender, grade, and region all had an impact on the intake of milk. Some results found that children in northern schools were more likely to consume milk than children in southern schools in the United States [29].

In addition, we investigated flavor preferences for flavored milk in Figure 1. Results indicated that people preferred strawberry flavor, milk flavor, chocolate flavor, wheat flavor, red date flavor, mango flavor, and yellow peach flavor, while raspberry flavor, red bean flavor, passion fruit flavor, and pineapple flavor were liked by fewer people. Taste had a significant impact on children’s flavored milk consumption, and a high association with brands and emotions was shown. Some studies discovered that 50% of Belgian children (8~13 years old) preferred chocolate flavor first and fruit flavors second [30] and removing the option of chocolate flavored milk significantly reduced intake of milk [31].

### 3.2. Composition and Content of Flavor Samples

In flavor samples, 285 flavor components were identified and the basic information of flavor compounds is shown in Appendix A. As shown in Figure 2a, the flavor samples mainly consisted of esters (32.17%), alcohols (11.19%), olefins (9.09%), aldehydes (8.39%), ketones (7.34%), aromatic compounds (4.20%), and pyrazines (3.85%). Various fruit flavors (68.2%) made up the majority of flavor samples, and studies showed that fruit volatile compounds were mostly composed of esters, alcohols, aldehydes, and ketones [32]. 

The detection rate was calculated by the ratio of flavor component detection times to the number of flavor samples. Compounds with a detection rate > 35% were as follows: methyl palmitate (90.91%) > ethyl butyrate (81.82%) = dipentene (81.82%) > ethyl laurate (77.27%) > γ-decalactone (72.73%) > isoamyl acetate (68.18%) = benzaldehyde (68.18%) > linalool (63.64%) = hexyl acetate (63.64%) > ethyl 2-methylbutyrate (54.55%) = peach aldehyde (54.55%) > ethyl phenylacetate (50.00%) = dodecanol (50.00%) > citrated acetone (45.45%) = nonanal (45.45%) > benzyl acetate (40.91%) = α-terpineol (40.91%) > etheyl octanoat (36.36%) = leaf alcohol (36.36%) = lauryl alcohol (36.36%) = ethyl caprate (36.36%). Methyl palmitate was the most common compound in flavors and fragrance [33]. Alphonso mango flavor (26.14%), pineapple flavor (54.4%), and passion fruit flavor (24.67%) were mainly composed of allyl hexanoate. Strawberry concentrate flavor (18.02%), golden mango flavor (37.49%), peach flavor (24.77%), and coconut flavor (16.18%) were mainly made up of γ-decalactone. Although different flavor samples shared many volatile compounds, each flavor had a distinctive aroma depending upon the volatile mixture, concentration, and perception threshold of individual volatile compounds (Figure 2c).

### 3.3. Composition Analysis of Seven Flavor Samples

Strawberry flavor, milk flavor, chocolate flavor, wheat flavor, red date flavor, mango flavor, and yellow peach flavor were analyzed, as shown in Figure 2b–d. A total of 168 compounds were identified, esters being the main components among the seven flavor samples. The strawberry flavor had the highest proportion of esters, as the most important category [34], including propyl decalactones, ethyl 2-methylbutyrates, ethyl butyrate, methyl cinnamate, phyllyl acetate, and ethyl caproate. These compounds were also detected in previous studies [35]. The PCA (Figure 2c) showed that mango and strawberry had the most similar composition. Mango had a higher percentage of ethers, anhydrides, alcohols, and olefins [36,37].

The chocolate flavor had a complex composition of aldehydes, pyrazines, alcohols, esters, ketons, furans, acids, and phenols [38]. According to this study, chocolate had the most aldehydes, while other studies discovered pyrazines were the major volatile and key odor compounds in chocolate flavor. It is possible that pyrazines produced by the Maillard reaction were the most important compounds that contributed to the final chocolate flavor [39]. The PCA diagram revealed that chocolate and wheat flavors had a similar composition. Wheat flavor had the largest amount of pyrazines and alkanes. According to the heat map (Figure 2d), wheat flavor mostly consisted of pyrazines, thiazoles, and furan compounds. Milk flavor had the highest proportion of alcohols relative to other flavor samples and was mainly composed of alkanes, phenols, and acids. Yellow peach flavor was mainly made up of esters, alcohols, aldehydes, and ketones. Additionally, yellow peach flavor had a higher content of peach aldehyde (26.26%), benzyl acetate (17.71%), and hexyl acetate (16.64%) which was considered key odorants influencing the flavor quality of peach fruit [40].

In the PCA analysis, the red date flavor had a unique flavor composition. The largest class of aroma-impact compounds was esters, including ethyl laurate, ethyl palmitate, methyl hydroxyacetate, and isopentyl acetate. Another important class of odor-active chemicals was aldehydes, and three aroma-impact aldehyde compounds were found in the samples. 5-methylfuranal, furfural, and benzaldehyde were among them [41]. Flavor is a complex mixture of volatile compounds, and the composition was specific to the species and variety of fruits [42]. 

### 3.4. The Screening of Flavor Concerned Component

The screening process related to the acute toxicity dosage grading standard in GB 15193.3-2014, and took LD 50 < 3000 as the basic screening criterion under the following two conditions: (I) detection rate greater than 10% (frequency of detection in all flavor samples) and LD 50 < 3000. (II) The detection rate was lower than 10%, the components of seven favorite flavor samples accounted for more than half of the total, and LD 50 < 3000. The fifteen flavor ingredients of concern were 2-methylpropanal, ethyl 3-methylbutyrate, 1-hexanol, allyl hexanoate, 2,3,5-trimethylpyrazine, furfural, benzaldehyde, linalool, 5-methylfurfural, benzyl acetate, methyl salicylate, benzenemethanol, maltol, methyleugenol, and phenol 2-methoxy-4-(2-propenyl)-, as shown in Table 3. The linear equation, R2, and detection limit of flavor components of concern are shown in Appendix A.

### 3.5. Quantitative Analysis of Flavor Concerned Components in Flavored Milk 

Fifteen components of concern were quantitatively analyzed in flavored milk samples and shown in Table 4. Benzenemethanol, 2,3,5-trimethylpyrazine, furfural, and benzaldehyde were all detected in 100% of flavored milk samples. Benzenemethanol is a colorless liquid with a mild pleasant aromatic odor naturally produced by fruits and teas. Pyrazines are nitrogen-containing heterocyclic compounds that contribute significantly to the flavor of various grilled, roasted, and similarly cooked foods, including baked potatoes, nuts, and meats [46]. Differences in furfural and benzaldehyde levels in flavored milk were generated by the Maillard reaction and the protein denaturation reaction. Benzaldehyde is an aromatic aldehyde bearing a single formyl group and an almond odor, and can be extracted naturally and is widely utilized in the production of aniline dyes, perfumes, flavorings, and medicines. In addition, the detection rates of 5-methylfurfural (96.4%), maltol (96.4%) and 1-hexanol (92.9%) were higher than 90%. Maltol is one of the byproducts of sugar degradation. There were differences in the content of flavor components of concern in various flavored milk brands, which were influenced by additive amount and manufacturing techniques. Maltol concentrations ranged from 0.83 μg kg^−1^ to 1682.11 μg kg^−1^. The maximum content of benzenemethanol (14,995.44 μg kg^−1^) was determined, which was significantly higher than 2,3,5-trimethylpyrazine (2387.18 μg kg^−1^), furfural (3840.42 μg kg^−1^), and linalool (4958.30 μg kg^−1^). 

### 3.6. Risk Exposure Assessment of Flavored Milk

The maximum EDI of different age groups was found to be considerably smaller than the ADI in the risk evaluation. The findings suggested that the flavor components had nothing exposure risks or health threats to the Chinese people. The detailed data is showed in Table 5. However, EDI values were different among age groups. Among people < 18 years old, EDI (0.023~20.27 μg kg^−1^, bw d^−1^) was much higher than other age groups, which was closely related to the high flavored milk consumption by teenagers. The EDIs of benzenemethanol (5.231~20.27 μg kg^−1^, bw d^−1^), furfural (1.34~5.19 μg kg^−1^, bw d^−1^), and linalool (1.73~6.70 μg kg^−1^, bw d^−1^) were much higher than other components. 

The PCI of different age groups was less than or significantly less than TTC. The results of risk assessment are shown in Table 6 and indicated that there was no exposure risk to human health. The results showed that PCI of 2,3,5-trimethylpyrazine, furfural, linalool, and benzenemethanol were the highest across different groups, particularly among people under 18 years old. The PCI of furfural (297.92 g p^−1^ d^−1^) and benzenemethanol (1163.26 g p^−1^ d^−1^) in males (>18 years old) was closed due to their toxicological concern thresholds of 540 and 1800, respectively. Therefore, children should pay closer attention to the consumption of flavored milk. The maximum daily consumption was estimated using the TTC of the concerned components. The daily maximum intake of three flavor components (2,3,5-trimethylpyrazine, furfural, benzenemethanol) was less than one box of milk (250 g). The highest linalool and maltol consumption was less than two boxes of milk (250 g). This might serve as a starting point for additional research into maximal exposure and utilization in flavored milk.

## 4. Conclusions

Through the investigation of the intake of flavored milk in different regions, ages, and demographics, it was found that males (<18 years old) in southwestern regions had the highest intake of flavored milk. At the same time, 285 components of different flavor components were determined, and 15 flavor components of concern were screened for risk assessment. Two risk assessments confirmed that Chinese residents’ intake of flavors in flavored milk was safe for their bodies. In addition, the maximum intake of 2,3,5-trimethylpyrazine (226.21 g), furfural (140.61 g), and benzenemethanol (120.04 g) was less than 250 g, which can provide a reference value for flavor additive amounts in milk. The result of per capita dairy product consumption in the questionnaire survey was a rough estimate and all the estimates were based on one-week records of flavored milk consumption. Although this may not represent the usual intake, it was enough for estimating people’s average intake.

## Figures and Tables

**Figure 1 foods-12-02151-f001:**
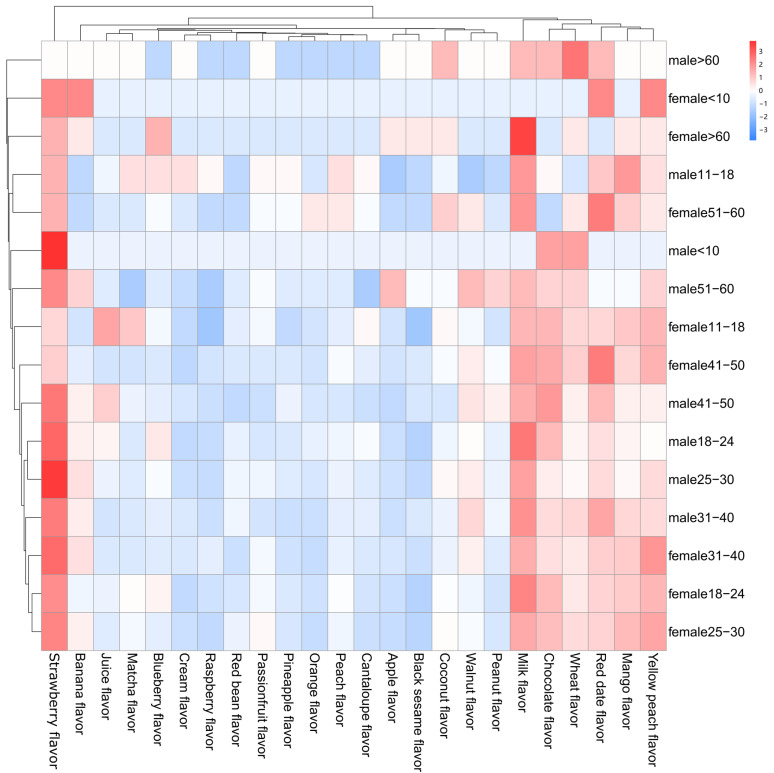
Heat map analysis of Chinese residents’ preference in milk flavors.

**Figure 2 foods-12-02151-f002:**
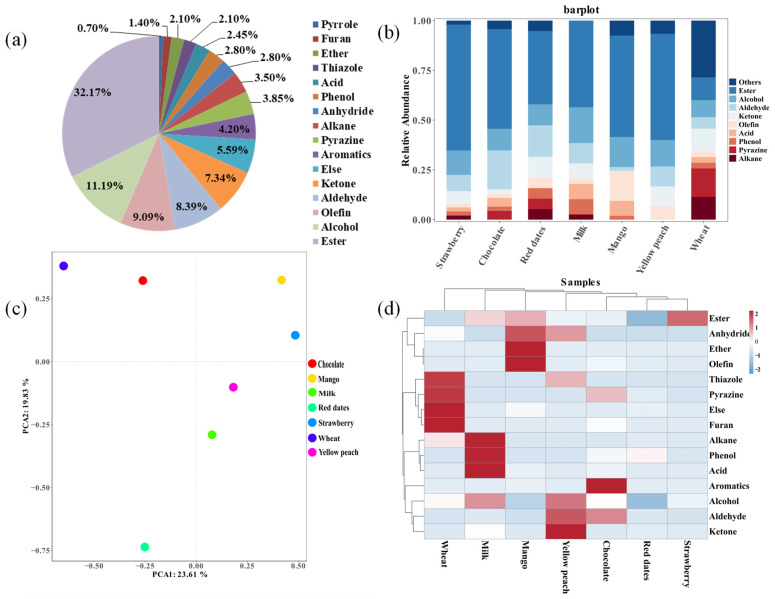
Visualized analysis of the volatile compounds in flavor samples: (**a**) content ratios in types of volatile compounds among all flavor samples; (**b**) bar plot diagram analysis of the compound types in seven flavor samples; (**c**) classification by PCA of the seven flavor samples; (**d**) cluster heat map analysis of the compound content of the different types.

**Table 1 foods-12-02151-t001:** The adding amount of flavor ingredients in different foods regulated by FEMA (mg kg^−1^).

Components	Dairy Products	Soft Drinks	Cold Drinks	Candies	Bakery Products	Liquor	Pudding	Gum Confectionary	Meat and Meat Sauces	Syrup	Chewing Gum	Jelly
2-methylpropanal	-	0.30	0.25~0.50	0.67	0.5~1.0	5	-	-	-	-	-	-
ethyl 3-methylbutyrate	-	4.90	7.50	29	27	-	5	80~430	-	-	-	-
1-hexanol	-	6.60	26	21	18	-	0.22~0.28	-	-	-	-	-
allyl hexanoate	-	7	11	32	25	-	22	-	-	-	-	-
2,3,5-trimethylpyrazine	1	5.00~10	-	5.00~10	5.00~10	-	-	-	2	-	-	-
furfural	-	4	13	12	17	10	0.80	45	-	30	-	-
benzaldehyde	-	36	42	120	110	50~60	160	840	-	-	-	-
linalool	-	2	3.60	8.40	9.60	-	2.30	0.80~90	40	-	-	-
5-methylfurfural	-	0.13	0.13	0.03~0.13	0.03	-	-	-	-	-	-	-
benzyl acetate	-	7.80	14	34	22	-	23	760	-	-	-	-
methyl salicylate	-	59	27	840	54	-	-	8400	-	200	-	-
benzenemethanol	-	15	160	47	220	-	-	-	-	-	1200	-
maltol	-	4.10	8.70	31	30	-	7.50	-	-	-	90	15
methyleugenol	-	10	4.80	11	13	-	-	-	-	-	-	52
phenol,2-methoxy-4-(2-propenyl)-	-	4.60	3.80	6.80	9	-	4	-	-	-	0.30	-

**Table 2 foods-12-02151-t002:** Age and regional grouping, average weight, daily consumption.

Gender	Age	Average Weight/kg *	Day per Capita Consumption/g *	Coefficient of Variation	Area	Average Weight/kg *	Day per Capita Consumption/g *	Coefficient of Variation
Male	<18	57.39 ± 10.95	77.57 ± 89.34	1.15	Eastern China	67.11 ± 9.24	53.08 ± 96.82	1.82
	18~24	65.89 ± 10.29	57.66 ± 89.59	1.55	Southern China	63.43 ± 9.97	50.86 ± 68.90	1.35
	25~30	70.09 ± 9.91	50.49 ± 135.40	2.68	Central China	68.99 ± 9.98	42.71 ± 52.44	1.23
	31~40	72.69 ± 9.28	37.90 ± 62.89	1.66	Northern China	71.21 ± 10.48	55.13 ± 133.73	24.19
	41~50	73.73 ± 9.48	59.21 ± 127.55	2.15	Northwestern China	69.19 ± 10.50	50.53 ± 69.15	1.37
	>51	71.61 ± 10.35	49.99 ± 64.02	1.28	Southwestern China	63.23 ± 9.41	50.06 ± 54.18	1.08
					Northeastern China	70.01 ± 11.80	64.33 ± 87.32	1.36
Female	<18	51.62 ± 7.05	55.14 ± 52.44	0.95	Eastern China	54.80 ± 8.50	42.54 ± 54.63	1.28
	18~24	53.33 ± 8.33	50.57 ± 56.66	1.12	Southern China	51.58 ± 7.94	45.21 ± 50.21	1.11
	25~30	54.87 ± 7.79	35.13 ± 46.75	1.33	Central China	54.53 ± 8.33	55.82 ± 57.14	1.02
	31~40	57.41 ± 9.81	20.03 ± 31.94	1.59	Northern China	55.89 ± 9.01	38.00 ± 52.23	1.38
	41~50	58.86 ± 7.67	35.32 ± 51.76	1.47	Northwestern China	55.17 ± 7.29	46.50 ± 55.75	1.20
	>51	60.10 ± 8.54	37.70 ± 55.60	1.47	Southwestern China	51.41 ± 7.27	42.14 ± 46.22	1.10
	*p*	***	***		Northeastern China	56.84 ± 9.41	36.12 ± 48.83	1.35
					*p*	***	0.03 **	

* Mean values (M) and standard deviation (SD). *** shows a significant difference between columns (*p* < 0.01). ** shows a difference between columns (*p* < 0.05).

**Table 3 foods-12-02151-t003:** Flavor concerned components information.

Number	RT/min	CAS Number	Name	Molecular Formula	Calculated RI	Library RI	LD50/mg/k	ADI * (μg kg^−1^, bw d^−1^)	TTC (μg kg^−1^, bw d^−1^)
1	6.559	78-84-2	2-methylpropanal	C_4_H_8_O	812	820	960	500	1800
2	14.629	108-64-5	ethyl 3-methylbutyrate	C_7_H_14_O_2_	1067	1082	1200	1500	1800
3	27.896	111-27-3	1-hexanol	C_6_H_14_O	1350	1355	720	1200	1800
4	28.936	123-68-2	allyl hexanoate	C_9_H_16_O_2_	1373	1360	218	130	540
5	30.277	14667-55-1	2,3,5-trimethylpyrazine	C_7_H_10_N_2_	1402	1402	806	500	540
6	33.159	98-01-1	furfural	C_5_H_4_O_2_	1468	1477	65	960	540
7	35.756	100-52-7	benzaldehyde	C_7_H_6_O	1530	1541	1300	5000	1800
8	36.379	78-70-6	linalool	C_10_H_18_O	1545	1549	2790	500	1800
9	37.753	620-02-0	5-methylfurfural	C_6_H_6_O_2_	1578	1588	2200	5000	540
10	43.739	140-11-4	benzyl acetate	C_9_H_10_O_2_	1736	1720	2490	5000	1800
11	45.34	119-36-8	methyl salicylate	C_8_H_8_O_3_	1786	1796	887	500	1800
12	47.966	100-51-6	benzenemethanol	C_7_H_8_O	1881	1890	1230	5000	1800
13	50.117	118-71-8	maltol	C_6_H_6_O_3_	1973	1984	1410	1000	540
14	51.074	93-15-2	methyleugenol	C_11_H_14_O_2_	2017	2013	810	5000	1800
15	54.152	97-53-0	phenol,2-methoxy-4-(2-propenyl)-	C_10_H_12_O_2_	2179	2185	1930	2500	1800

* The ADI (μg kg^−1^, bw d^−1^) of allyl hexanoate, benzaldehyde, linalool, benzyl acetate, methyl salicylate, benzenemethanol, maltol, phenol 2-methoxy-4-(2-propenyl)- was collected by JECFA and got from Pubchem (https://pubchem.ncbi.nlm.nih.gov/ accessed on 9 March 2023). ADI of 2-methylpropanal was provided by the European Food Safety Authority. Furfural [43], ethyl 3-methylbutyrate [44], and 2,3,5-trimethylpyrazine [45], were obtained from relevant studies. ADI of 5-methylfurfural and methyleugenol were given by the Communauté européenne.

**Table 4 foods-12-02151-t004:** Quantitative analysis results of 28 flavored milk samples.

Number	Compounds	Minimum	Median	Maximum	Mean ± SD	Detections	Detection Rate
		(μg kg^−1^)					
1	2-methylpropanal	7.64	32.15	679.44	135.07 ± 209.71	20	71.40
2	ethyl 3-methylbutyrate	1.11	1.20	21.74	3.55 ± 6.43	9	32.10
3	1-hexanol	4.60	6.12	179.83	17.23 ± 34.86	26	92.90
4	allyl hexanoate	13.98	16.24	270.84	52.21 ± 88.41	14	50.00
5	2,3,5-trimethylpyrazine	13.72	14.74	2387.18	162.19 ± 488.82	28	100.00
6	furfural	12.69	16.61	3840.42	158.71 ± 708.66	28	100.00
7	benzaldehyde	15.05	26.10	931.77	98.07 ± 208.00	28	100.00
8	linalool	11.11	11.43	4958.30	218.77 ± 968.39	25	89.30
9	5-methylfurfural	12.47	13.37	1050.97	66.38 ± 200.66	27	96.40
10	benzyl acetate	9.87	17.35	643.09	117.37 ± 203.89	16	57.10
11	methyl salicylate	11.51	20.62	27.30	20.01 ± 5.95	4	14.30
12	benzenemethanol	20.61	28.96	14,995.44	773.61 ± 2890.17	28	100.00
13	maltol	0.83	60.14	1682.11	353.91 ± 531.51	27	96.40
14	methyleugenol	24.12	24.55	24.82	24.51 ± 0.27	4	14.30
15	phenol,2-methoxy-4-(2-propenyl)-	23.69	24.30	1418.60	129.61 ± 331.14	17	60.70

**Table 5 foods-12-02151-t005:** Comparison of maximum EDI and ADI between different consumer groups.

Compounds	Maximum EDI (μg kg^−1^, bw d^−1^)												ADI (μg kg^−1^, bw d^−1^)
	Male/Years						Female/Years						
	<18	18–24	25–30	31–40	41–50	>50	<18	18–24	25–30	31–40	41–50	>50	
2-methylpropanal	0.92	0.60	0.49	0.35	0.55	0.47	0.73	0.64	0.44	0.24	0.41	0.43	500
ethyl 3-methylbutyrate	0.03	0.02	0.02	0.01	0.02	0.02	0.02	0.02	0.01	0.01	0.01	0.01	1500
1-hexanol	0.24	0.16	0.13	0.09	0.14	0.13	0.19	0.17	0.12	0.06	0.11	0.11	1200
allyl hexanoate	0.37	0.24	0.20	0.14	0.22	0.19	0.29	0.26	0.17	0.09	0.16	0.17	130
2,3,5-trimethylpyrazine	3.23	2.09	1.72	1.25	1.92	1.67	2.55	2.26	1.53	0.83	1.43	1.50	500
furfural	5.19	3.36	2.77	2.00	3.08	2.68	4.10	3.64	2.46	1.34	2.30	2.41	960
benzaldehyde	1.26	0.82	0.67	0.49	0.75	0.65	1.00	0.88	0.60	0.33	0.56	0.58	5000
linalool	6.70	4.34	3.57	2.59	3.98	3.46	5.30	4.70	3.18	1.73	2.98	3.11	500
5-methylfurfural	1.42	0.92	0.76	0.55	0.84	0.73	1.12	1.00	0.67	0.37	0.63	0.66	5000
benzyl acetate	0.87	0.56	0.46	0.34	0.52	0.45	0.69	0.61	0.41	0.22	0.39	0.40	5000
methyl salicylate	0.04	0.02	0.02	0.01	0.02	0.02	0.03	0.03	0.02	0.01	0.02	0.02	500
benzenemethanol	20.27	13.12	10.80	7.82	12.04	10.47	16.02	14.22	9.60	5.23	9.00	9.41	5000
maltol	2.27	1.47	1.21	0.88	1.35	1.17	1.80	1.60	1.08	0.59	1.01	1.06	1000
methyleugenol	0.03	0.02	0.02	0.01	0.02	0.02	0.03	0.02	0.02	0.01	0.02	0.02	5000
phenol,2-methoxy-4-(2-propenyl)-	1.92	1.24	1.02	0.74	1.14	0.99	1.52	1.35	0.91	0.50	0.85	0.89	2500

**Table 6 foods-12-02151-t006:** Comparison of maximum PCI and TTC between different age groups.

Compounds	Maximum PCI (μg p^−1^ d^−1^)												TTC (μg kg^−1^, bw d^−1^)	PCI/TTC > 1	Max per Capita Daily Intake/g	Max Box/ 250 g
	Male/Years						Female/Years									
	<18	18–24	25–30	31–40	41–50	>50	<18	18–24	25–30	31–40	41–50	>50				
2-methylpropanal	52.71	39.18	34.31	25.75	40.23	33.97	37.47	34.36	23.87	13.61	24.00	25.61	1800	NO	2649.24	10.60
ethyl 3-methylbutyrate	1.69	1.25	1.10	0.82	1.29	1.09	1.20	1.10	0.76	0.44	0.77	0.82	1800	NO	82,796.69	331.20
1-hexanol	13.95	10.37	9.08	6.82	10.65	8.99	9.92	9.09	6.32	3.60	6.35	6.78	1800	NO	10,009.45	40.00
allyl hexanoate	21.01	15.62	13.68	10.27	16.04	13.54	14.94	13.70	9.52	5.42	9.57	10.21	540	NO	1993.80	8.00
2,3,5-trimethylpyrazine	185.18	137.65	120.54	90.48	141.35	119.34	131.64	89.98	83.87	47.80	84.31	89.98	540	NO	226.21	0.90
furfural	297.92	221.44	193.92	145.56	227.40	191.98	211.77	194.21	134.92	76.90	135.63	144.76	540	NO	140.61	0.60
benzaldehyde	72.28	53.73	47.05	35.32	55.17	46.58	51.38	47.12	32.74	18.66	32.91	35.12	1800	NO	1931.81	7.70
linalool	384.64	285.90	250.36	187.93	293.60	247.87	273.42	250.74	174.20	99.29	175.11	186.90	1800	NO	363.03	1.50
5-methylfurfural	81.53	60.60	53.07	39.84	62.23	52.54	57.95	53.15	36.92	21.05	37.12	39.62	540	NO	513.81	2.10
benzyl acetate	49.89	37.08	32.47	24.38	38.08	32.15	35.46	32.52	22.59	12.88	22.71	24.24	1800	NO	2798.99	11.20
methyl salicylate	2.12	1.57	1.38	1.04	1.62	1.37	1.51	1.38	0.96	0.55	0.96	1.03	1800	NO	65,934.07	263.70
benzenemethanol	1163.26	864.64	757.17	568.37	887.93	749.63	826.90	758.31	526.83	300.28	529.60	565.25	1800	NO	120.04	0.50
maltol	130.49	96.99	84.94	63.76	99.60	84.09	92.76	85.06	59.10	33.68	59.41	63.41	540	NO	321.03	1.30
methyleugenol	1.93	1.43	1.25	0.94	1.47	1.24	1.37	1.26	0.87	0.50	0.88	0.94	1800	NO	72,522.16	290.10
phenol,2-methoxy-4-(2-propenyl)-	110.05	81.80	71.63	53.77	84.00	70.92	78.23	71.74	49.84	28.41	50.10	53.47	1800	NO	1268.86	5.10

## Data Availability

Data is contained within the article or Appendix A.

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
