# Peer review of "Determination and Risk Assessment of Flavor Components in Flavored Milk"

_foods, 2023, doi:10.3390/foods12112151_

Round 1
Reviewer 1 Report
Title: Determination and risk assessment of flavor components in flavored milk.
In this article, the authors make an attempt to assess the risk of exposure to substances present in flavored milk drinks. This assessment is carried out in a sample of the Chinese population and is based on the collection of data on the consumption of the products from 2108 responses to a questionnaire. From this questionnaire, demographic data, respondents' consumption of flavored milk drinks, and preferences for the type of drink were collected. Furthermore, the flavored milk drinks were analyzed and the volatile substances responsible for the aroma were determined.
Finally, a risk assessment was carried out for exposure to the substances determined by estimating exposure by the population and comparing this exposure estimate with established ADI and TTC data.
The conclusions indicate that for none of the 15 substances determined, in no case did the exposure exceed the ADI and TTC values.
The article addresses an interesting research topic that has not been addressed previously. The paper is well structured, although some general issues need to be addressed.
-The first question has to do with how the information on the consumption of the products under study was obtained. The article mentions that a questionnaire was used but does not indicate the type of questionnaire used or whether it has been previously validated. The measurement of intake in any population is difficult to perform and is considered one of the major methodological problems in nutritional epidemiology and risk estimation. Please expand the information provided on this questionnaire and, if possible, include the model of the questionnaire used as supplementary information.
-The second question is related to the data obtained with the previous questionnaire. Table 2 shows this information. The data are shown as the mean and its standard deviation. It is interesting to point out that food consumption data do not usually fit a normal distribution, therefore, it would be interesting for the authors to discuss this issue and assess which value to take to perform the risk assessment (the mode, the mean, or the median). On the other hand, the significant figures with which the estimators of the most frequent consumption and its variability are provided should be reviewed.
-The third point to be clarified concerns the selection of the compounds to be studied. The selection was based on two aspects: the first is the frequency of detection, and the second is the LD50, i.e. an index of acute toxicity. Since we are dealing with substances present in a type of food that is consumed with a certain frequency, would it not have been more logical to choose those with a lower ADI value? That is to say, to use the ADI value as a criterion and not the LD50, since, in addition, the risk assessment that the authors are trying to make is not for acute exposure but for chronic exposure. Also, the units of the ADI and TTC data should be included in Table 3, please.
-The fourth point to comment is related to the experimental procedure for the determination of the aromas. The authors indicate that the method of reference 16 has been followed. It would be appreciated if the authors could provide more information on the analytical methodology used, LOD, LOQ, regression coefficient, and linear interval in Table 4. HS-SPME is a technique that presents a certain complexity and it would be interesting to have this information available.
-The fifth aspect to be addressed is the fact that a deterministic risk assessment has been carried out by taking the mean value as the food consumption data and the maximum detected value as the substance concentration data. The fact that it is a deterministic procedure should be mentioned both in the title and throughout the article. On the other hand, the analysis could be extended by taking into account the consumption data of the population considered as high consumers (i.e. with a consumption equal to 95% of the maximum consumption value), as well as those with a consumption equivalent to the mode or median.
Other minor issues are as follows:
What is meant by increasing solid fat intake (page 4, second paragraph, second line starting from the end).
As it has been said, please revise the significant figures of values shown throughout the text.
Reviewer 2 Report
References Numbers 4, 6, 17 and 22 are very old and need replacing modern references
Some words need revision
Reviewer 3 Report
Please mention table 1, in the main text before its presentation.
In general the manuscript is very interesting, well constructed and the experiment was deisgned professionally.
I have no further comments for imporvement.
Best regards
